# Genotype and Phenotype Influence the Personal Response to Prostaglandin Analogues and Beta-Blockers in Spanish Glaucoma and Ocular Hypertension Patients

**DOI:** 10.3390/ijms24032093

**Published:** 2023-01-20

**Authors:** Valeria Opazo-Toro, Virginia Fortuna, Wladimiro Jiménez, Marta Pazos López, María Jesús Muniesa Royo, Néstor Ventura-Abreu, Mercè Brunet, Elena Milla

**Affiliations:** 1Glaucoma Unit, Instituto Oftalmologico Integral, C/María Auxiliadora 25, 08017 Barcelona, Spain; 2Pharmacology and Toxicology Laboratory, Biochemistry and Molecular Genetics Service, Biomedical Diagnostic Center, Hospital Clinic Barcelona, University of Barcelona, 08007 Barcelona, Spain; 3Biochemistry and Molecular Genetics Service, Center for Biomedical Diagnosis, Hospital Clinic Barcelona, 08036 Barcelona, Spain; 4August Pí i Sunyer Research Institute (IDIBAPS), University of Barcelona, 08036 Barcelona, Spain; 5Glaucoma Unit, Institut Clínic d’Oftalmologia, Hospital Clínic, 08036 Barcelona, Spain; 6Glaucoma Unit, Hospital Sagrat Cor. Barcelona, 08029 Barcelona, Spain

**Keywords:** genotype, beta-blockers, prostaglandin analogues, glaucoma, personalized medicine, pharmacogenetics, prostaglandin-F2α receptor, CYP, drug action, single nucleotide polymorphism

## Abstract

Analysis of the genotype that predicts the phenotypic characteristics of a cohort of glaucoma and ocular hypertension patients, and the correlation with their personal pharmacological response to beta-blockers (BB) and prostaglandin analogues (PGA). Prospective study that included 139 eyes from 72 patients under BB and/or PGA treatment, and in some cases other types of ocular hypotensive treatments. Five single-nucleotide polymorphisms were genotyped by real-time PCR assays: prostaglandin-F2α receptor (rs3766355, rs3753380); cytochrome-P450 2D6 (rs16947, rs769258); and beta-2-adrenergic receptor (rs1042714). Other studied variables were mean deviation (MD) of visual field, previous ocular interventions, medical treatment, baseline (bIOP), and treated intraocular pressure (tIOP). From a total of 139 eyes, 71 (51.1%) were left eyes. The main diagnosis was primary open angle glaucoma (66.2%). A total of 57 (41%) eyes were under three or more medications (PGA + BB + other) and, additionally, 57 eyes (41%) had had some kind of glaucoma surgery. The mean bIOP and tIOP were 26.55 ± 8.19 and 21.01 ± 5.54 mmHg, respectively. Significant differences in tIOP were found between heterozygous (HT) (21.07 ± 0.607 mmHg) and homozygous (HM) (20.98 ± 0.639 mmHg) rs3766355 with respect to wildtype individuals (16 ± 1.08 mmHg) (*p* = 0.031). The MD values presented significant differences between wildtype rs3766355 (−2 ± 2.2 dB), HT (−3.87 ± 4 dB), and HM carriers (−9.37 ± 9.51 dB) (*p* = 0.009). Significant differences were also observed between the MD in wildtype rs3753380 (−6.1 ± 8.67 dB), HT (−9.02 ± 8.63 dB), and HM carriers (−9.51 ± 7.44 dB) (*p* = 0.017). Patients carrying the variant rs3766355 in HM or HT presented clinically-significantly higher tIOP than wildtype patients. Additionally, some differences in MD were found in rs3766355 and rs3753380 carriers, and the more alleles that were affected, the worse the MD value, meaning greater severity of the glaucoma. Poor response to treatment and more visual field damage may be associated with being a carrier of these mutated alleles.

## 1. Introduction

Glaucoma is the leading cause of irreversible blindness [1]. To prevent the progression of the glaucomatous defect, the treatment that has shown the greatest effectiveness is to decrease intraocular pressure (IOP) [2,3]. Starting an early and effective ocular hypotensive treatment is crucial to slowing glaucoma progression. In the treatment of glaucoma, different kinds of ocular hypotensive drugs are used nowadays to lower IOP, such as prostaglandin analogues (PGA), beta-blockers (BB), alpha2-agonists, carbonic anhydrase inhibitors, parasympathomimetics, and osmotics, among others [4]. According to different glaucoma guidelines, the most widely-used drugs in the treatment of glaucoma belong to the group PGA, which includes latanoprost and BB such as timolol maleate [4,5]. The rates of response to the ocular hypotensive drugs vary among populations but can be as high as 28% for timolol or 18% for latanoprost [6]. Further, a substantial number of patients present significant side effects, especially with beta-blockers [4].

The pharmacological group of PGA has two subgroups: prostanoids (bimatoprost) and prostaglandin analogues (latanoprost, tafluprost, and travoprost) [7]). These drugs decrease IOP by increasing uveoscleral flow by reducing resistance to the outflow of aqueous humor (HA). These drugs reduce IOP by 25–35% [4]. Latanoprost (the first molecule developed in the PGA family) is a prodrug of prostaglandin F2 Alpha that, when passing through the cornea, is hydrolyzed by a corneal esterase and becomes biologically active, binding with high affinity to the prostaglandin F2 Alpha receptor (PTGFR) [4,8]. The prostaglandin F receptor binds to and mediates the biological actions of Prostaglandin F_2α_, and is encoded in humans by the *PTGFR* gene. In human beings, the expression of the PTGFR protein had been detected in the corneal epithelium, ciliary epithelium, the circular portion of ciliary muscle, and iris stromal and smooth muscle cells [9]. 

FP receptors activate phosphatidylinositol metabolism through G-coupled proteins, resulting in the increase in intracellular free calcium concentrations and the modulation of various signaling cascades. PGF2α and prostaglandin FP agonists decrease IOP by increasing the uveoscleral outflow via an unconventional pathway. PGAs could activate prostaglandin receptors in ciliary muscle, iris root, and sclera. The possible mechanisms of PGAs include the relaxation of ciliary smooth muscles, the alteration of cytoskeletal, and the remodeling of the extracellular matrix of the uveoscleral pathway [10]. Although often broadly categorized into the PGA class, bimatoprost is actually a synthetic prostamide, which is structurally distinct from other prostaglandin analogues [11]. 

In contrast to the acidic prostaglandins, prostamides lack a carboxylic acid group in their chemical structure, rendering them neutral in solution. Several studies have suggested the presence of unique prostamide-sensitive receptors that differ from the receptors of other prostaglandin analogues. This has been subsequently supported by the discovery of prostamide antagonists. However, to date, no receptor that is unique for bimatoprost has yet to be cloned, and this point remains controversial. After topical instillation, a significant quantity of intact bimatoprost has been found in the ciliary body. Bimatoprost has also been found in its hydrolyzed free-acid form in the AH, suggesting that it enters the anterior chamber via the cornea as a prodrug as well [12].

The beta receptor antagonist or beta-blocker drugs are widely used in medicine. At the ocular level, the non-selective ones stand out, such as Timolol, Levobunolol, Methylpranolol, and Carteolol. As ocular hypotensive medications, they reduce IOP between 20–25%, decreasing AH production and reducing cAMP production at the level of the ciliary epithelium. It has been observed that between 10–20% of patients do not respond to this drug. On the other hand, systemic side effects of beta-blockers such as bronchospasm, bradycardia, and heart block are known. Betaxolol is a relatively cardioselective β-adrenoceptor-blocking drug, with no partial agonist (intrinsic sympathomimetic) activity and weak membrane-stabilizing (local anesthetic) activity. Betaxolol selectively and competitively binds to and blocks beta-1 (β1) adrenergic receptors in the heart, thereby decreasing cardiac contractility and rate. This leads to a reduction in cardiac output and lowers blood pressure [4,8,13].

An individual reaction to a specific drug is influenced by multifactorial agents including environmental, systemic, and genetic factors. In most cases, the reactions of a group of individuals are of the Gaussian type with non- to low responders at the lower end of the curve and high to ultra-high responders at the upper end of the curve. Different genes have been described to be involved in the response and metabolism of drugs [14,15,16,17].

Among the genes involved in the response to prostaglandin analogs are those that encode the PTGFR. Various single nucleotide polymorphisms (SNPs) have been described to be associated with a different response to latanoprost, namely rs3766355 and rs3753380. It was observed that patients carrying the “mutated” allele (homozygous or heterozygous) had an increased bIOP and responded worse to treatment with latanoprost, compared to those carrying the homozygous “wildtype” allele. These SNPs caused a low response by producing an alteration in the *PTGFR* expression [18,19,20,21]. However a few studies have shown no correlation between those variants and the ocular response to latanoprost [22].

The pharmacological activity of BB in the body is influenced in part by beta adrenergic receptors whose variants may influence the personal response to treatment, as well as the metabolization phenotype for the biotransformation of these therapeutic drugs that is carried out by cytochrome p450 in the liver, which determines plasma concentration [23]. The beta adrenergic receptors, classically called β1, β2, and β3, are encoded in the beta adrenergic receptor (ADRB) genes 1, 2, and 3 (chromosome 10, 5, and 8, respectively). They are widely distributed in the body and their activity is mediated by G proteins. ADRB2 is the one found in the greatest quantities in ocular tissues at the level of the ciliary body, optic nerve, and trabecular meshwork; however, it is also expressed in the heart and the adipose tissue [8,13]. Polymorphisms (SNPs) have been described for ADRB genes that are associated with altered receptor function and, therefore, research indicating that they are responsible for low or non-response towards BB has been published. The Arg389Gly polymorphism in *ADRB1* has been observed to be associated with an increased bIOP and a greater response to topical treatment with betaxolol. Another polymorphism described is that associated with *ADRB2* Gln27Glu (rs1042714), which, in the form of greater homozygous, decreases at least 20% IOP with treatment compared to the heterozygous case that is more resistant to treatment [13,24]. 

The metabolism of beta-blockers is in charge of the cytochrome-p450-2D6 (CYP2D6) enzyme, which corresponds to 1–5% CYP in the liver, but which is responsible for metabolizing 25% of the drugs most used in medicine (antidepressants, antipsychotics, opioids, antiarrhythmics, etc). Likewise, this enzyme can be induced and inhibited by some drugs such as fluoxetine, which must be taken into account in polymedicated patients since it can increase the side effects of the drugs or prevent patients from reaching the therapeutic range [25]. The *CYP2D6* gene is encoded on chromosome 22q13.2, more than 70 alleles, and 130 SNPs and/or short insertion or deletion polymorphisms have been described, and four phenotypic groups have been identified according to the number of functioning alleles—slow, intermediate, extensive, and ultrafast metabolizers—causing a different clinical response [26].

To note, the interest in pharmacogenetics clinical implementation has grown significantly in the last decade. Pharmacogenetics influences drug pharmacokinetics and pharmacodynamics and seeks to enable the selection of the right drug and starting dose for each treated patient to improve clinical outcomes. Knowledge of genetic contributors to variable drug response for current therapeutic drugs has also expanded dramatically, such that the evidence now supports the clinical use of genotype data to guide pharmacological treatment in a variety of therapeutic areas [27], and, in the near future, glaucoma will probably also benefit from this technology. Importantly, guidelines for genotype-based selection of the initial drug dose have been published for some specific drugs by some expert working groups. The Clinical Pharmacogenetics Implementation Consortium (CPIC) (https://cpicpgx.org/, accessed on 30 June 2022), The Dutch Pharmacogenetics Working Group (DPWG) (https://www.pharmgkb.org/, accessed on 30 June 2022), and ref. [28], among others, provide some recommendations in an attempt to facilitate the achievement of personalized treatment. 

The aim of our study was to genetically test a cohort of glaucoma and ocular hypertension (OHT) patients treated with PGA, BB, or a combination of both drugs and correlate the genetic findings with the clinical response verified in terms of IOP control and ancillary tests stabilization.

## 2. Results

### 2.1. Clinical Features 

The demographic and clinical features of the patients are shown in Table 1. 

From a total of 72 patients, we included 139 eyes, 71 (51.1%) were left eyes. Mean age was 63.99 ± 11.99 years, with 63.88% of the patients being older than 60 years of age (46 from 72). A total of 55.5% (40 from 72) of the patients corresponded to female. The main diagnosis was primary open angle glaucoma (66.2%) and the second was ocular hypertension (12.9%). Of the 139 eyes, 63 (45.3%) did not have a clinical record of surgery, 57 (41%) eyes had required glaucoma surgery, and 19 (13.7) had required some other kind of ophthalmological surgery. Of the total population, 38 (27.4%) eyes were under monotherapy (6.5% BB alone and 20.9%PGA alone), and 44 (31.6%) eyes were under double therapy (BB and PGA 24 (17.3%); BB and other treatment 18 (12.95); PGA and other treatment 2 (1.4)); 57 (41%) eyes were under three or more medications (BB plus PGA and other treatment). Systemic side effects to ocular medication such as bradycardia or dyspnea were reported in 16 patients (8.33%) that were being treated with BB. When we studied the concomitant use of systemic treatments in those patients, 17 (23.6%)) corresponded to patients with systemic treatments that were metabolized by CYP2D6 such as escitalopram, quetiapine, or aripiprazole. Of a total of six patients with adverse effects to BB, three patients were under systemic medication metabolized by CYP2D6. In all of these cases, a change in the therapeutic regimen was performed, switching to another ocular hypotensive drug, or laser trabeculoplasty or surgery, which were effective measures to eliminate the symptoms. 

The mean and standard deviation for bIOP and tIOP were 26.55 ± 8.19 and 21.01 ± 5.54 mmHg, respectively. The mean defect (MD) in the visual fields of our sample was −7.59 ± 8.63. More details are specified in Table 2.

### 2.2. Genetic Features

The frequencies of the SNPs in our population are described as follows: 

For the *PTGFR* gene and SNP rs3766355: 5.55% (4 patients) of the cases were Wildtype (WT), 23.61% (17 patients) were heterozygous (HT), and 70.83% (51 patients) were homozygous (HM).

For the PTGFR gene and SNP rs3753380: 47.22% (34 patients) were WT, 48.61% (35 patients) were HT, and 4.16% (3 patients) were HM. 

For the *ADBR2* gene and SNP rs1042714: 30.55% (22 patients) were WT, 31.9% (23 patients) were HT, and 37.55% (27 patients) were HM. For the *CYP2D6* gene and SNP rs16947: 40.9% (29 patients) were WT, 46.4% (33 patients) were HT, and 12.7% (9 patients) were HM. For the *CYP2D6* gene and SNP rs769258: 85.9% (61 patients) were WT, and 14.1% (10 patients) were HT. 

For *PTGFR* gene, we found significant differences between being a carrier of SNP rs3753380 and visual field MD, as HM individuals presented lower MD values (−9.51 ± 7.44773 dB) than HT (−9.02 ± 8.63445 dB) and WT (−6.10 ± 8.67477 dB), with *p* = 0.017. Further, being a carrier of rs3766355 was related to presenting with a lower visual field MD for HM (−9.37 ± 9.51008 dB) than HT (−3.87 ± 4.0248 dB) and WT (−2.07 ± 2.20883 dB), with *p* = 0.009. Additionally, this SNP presented significant differences in tIOP among carriers: WT (16 ± 2.16 mmHg), HT (21.70 ± 3.486 mmHg), and HM (20.98 ± 6.125 mmHg), with *p* = 0.031. 

Finally we did not observe statistically significant differences in the levels of bIOP, tIOP, nor MD for the SNPs rs769258 and rs16947 of the *CYP2D6* gene between HM, HT, or WT individuals. For the *ADBR2* gene rs1042714, no differences among groups were found either.

See Table 3 for details.

We found statistically significant differences after separating PGA patients between prostanoids and prostaglandin analogues. The prostanoids group bIOP was 30.43 ± 8.394 mmHg, the prostaglandin analogues group bIOP was 25.84 ± 8.086 mmHg, and the non-PGA bIOP was 25.99 ± 9.680 mmHg (*p* = 0.044). The tIOP in non-PGA eyes was 18.96 ± 4.573 mmHg, the prostaglandin analogues tIOP was 22.33 ± 4.597 mmHg, and the prostanoids tIOP was 20.29 ± 4.611 mmHg with *p* = 0.020

## 3. Discussion

This is the first time, to our knowledge, that a pharmacogenetics real life study in glaucoma patients has been presented. There are other previous studies about the pharmacogenetic profile of patients under ocular hypotensive therapy [21,29]; however, we believe that our study is the closest to usual ophthalmological practice since patients with different therapeutic regimens have been included and not only those in monotherapy, since this situation is the most common in a glaucoma consultation. Glaucoma is a potentially blinding disease and lowering IOP with hypotensive topical drugs is mandatory to slow the progression of the pathology. The most commonly-used drugs to control this disease are PGAs and BBs. However, it is known that a significant proportion of patients are clinically non-responders to these drugs. The rate of global non-responders differs among several studies and populations [30,31]. Rossetti et al. reported a low rate of non-responders (4.1%) versus the Bimatoprost/Latanoprost Study Group that found a significant proportion of non-responders to latanoprost (51.5%) [32].

In our study, we tested the rate of IOP decrease in patients affected by glaucoma or OHT by comparing bIOP and tIOP, the latter being measured not before the third month of treatment to avoid the bias caused by late responsiveness. In some studies, this fact has been overlooked, as in the one by Zhang P. et al. [21]. 

Different SNPs in the *PTGFR* receptor and the *ADRB* receptors have been described, causing a lack of response towards PGAs and BB, respectively [13,18,19,20,21,24].

In this study, we found that patients carrying *PTGFR* rs3766355, both in the HT or HM variant, presented a statistically-significantly higher tIOP after treatment with PGAs, which translates to a lower degree of therapeutic response. SNP rs3753380 was described previously by Sakurai et al. [19] as a risk factor for a poor response to latanoprost in a Japanese population and was also found in Caucasian non-responder patients by Ussa et al. [29], among others. We did not find a positive correlation between this variant and the ocular response to latanoprost.

In the study by Zhang P. et al., A correlation was found between rs3766355 of the *PTGFR* gene and response to latanoprost, and, although there was no correlation between rs3766355 with response to latanoprost on day 7, it showed a trend toward significance, and was statistically significant on day 30 [21]. Polymorphism rs3766355 of the *PTGFR* gene seems to be the most prevalent variant associated with a difference in response to PGAs, as described by other authors [18,19,20,21].

In the present study, patients presented an association between the values of visual field MD and being a carrier of *PTGFR* rs3766355 and rs3753380. Patients with these SNPs, both HT and HM, showed a greater alteration in the visual field, represented by a higher MD value. These values correspond to a higher degree of severity of the disease. To our knowledge, this is the first time in the literature that the association between pharmacogenetic profile in glaucoma and the visual field characteristics has been described. A worse response to medication contributes to a greater progression that translates into higher values of MD in the visual field.

When we differentiated PGA and prostanoid patients, we found statistically significant differences in the response between both drugs, with tIOP for prostanoid patients being lower than the tIOP for the other subgroup. It is known that prostanoids such as bimatoprost might be more effective at decreasing IOP than PGAS [33]. The clinical trial by Konstas A. et al. [34], which compared Latanoprost 0.005% versus bimatoprost 0.03% in primary open-angle glaucoma patients, indicated that the 24-h diurnal IOP was statistically lower in POAG with bimatoprost, compared with latanoprost. In addition, The Canadian Glaucoma Study confirmed the importance of intraocular pressure in glaucoma progression and that even the decrease of 1 mmHg in IOP could have consequences upon the optic disk deterioration [35]. This could explain why, in our study, which reflects real life clinical practice, the basal pressure of the patients who started treatment with bimatoprost was higher than that of the rest of the prostaglandin analogues, as, when faced with a patient with very high initial pressures, an attempt is made to find the medication that most effectively reduces IOP. Therefore, when the target IOP is lower or a more drastic IOP decrease is needed, as in advanced optic nerve damage or younger patients, in real life practice, bimatoprost is often used as an initial treatment. 

Among the risk factors for the development and progression of glaucoma is age. When analyzing our patients by age groups older and younger than 60 years old, no difference was observed in the frequency of carriers of the *PTGFR* gene of the SNPs mutation, rs3766355 and rs3753380. We did find differences in the response to medications where a worse response to treatment with beta-blockers and prostaglandin analogues was observed in those over 60 years of age; however, no difference was found in the mean pressures obtained after treatment in the groups of over and under 60 years.

Regarding the other genetic variants, we did not find statistical significance for the rest of the studied SNPs, neither in the IOP rate of decrease nor in the MD values. 

We also performed some investigations on the pharmacokinetic behaviour of BB in our population and tested two known SNPs of the CYP2D6 gene associated with variable responses to drugs. On one hand, SNP rs16947 may confer susceptibility to timolol-induced bradycardia. According to the study by Yuan H., patients with the CC genotype were unlikely to suffer from timolol-induced bradycardia, whereas those with the TT genotype were found to suffer from timolol-induced bradycardia [36]. On the other hand, SNP rs769258 is known to be present significantly more frequently in Caucasian ultrarapid metabolizer subjects than in extensive metabolizer subjects [37]. The identification of subjects with ultrarapid metabolic capacity is of potential clinical value. When patients do not respond to generally recommended doses and lower than expected plasma concentrations in relation to dose are measured, it is important to be able to distinguish between high metabolic capacity and non-compliance. Ultrarapid metabolism mediated by CYP2D6 is often due to the inheritance of alleles with duplicated or amplified functional *CYP2D6* genes; however, the frequency of the *CYP2D6* duplication/amplification alleles differs widely between ethnic groups. We did not find any correlation between these mentioned variants and the presence of cardiac or pulmonary side effects that are secondary to BB. 

Beta-blockers are a major component of therapy in numerous cardiovascular diseases, which have a higher incidence in older people. Along with polypharmacy and neurocognitive decline, potentially limiting reliable medication adherence, older patients may be more sensitive to the hypotensive and bradycardic effects of beta-blockers [38]. As glaucoma increases with age, topical BB must be administered with high precaution in these patients. In general, many β-blockers are metabolized by a highly polymorphic drug-metabolizing enzyme, CYP2D6. A total of 122 SNPs and/or short insertion/deletion polymorphisms have been reported within the *CYP2D6* genomic locus in human populations, resulting in at least 70 unique CYP2D6 haplotypes. The study by Nieminen T. et al. demonstrated that *CYP2D6* poor metabolizers presented altered serum kinetics following administration of the aqueous formulation of timolol (0.5% aqueous timolol) with higher rates of bradycardia, although not for the hydrogel formulation (0.1% timolol hydrogel). These findings suggest that, in the absence of knowledge regarding a patient’s *CYP2D6* genotype, it may be safer to prescribe the formulation (hydrogel) with the less variable kinetic profile [23]. In our study, half of the patients with adverse drugs effects were treated with concomitant systemic medication metabolized by CYP2D6, and they presented some systemic side effects such as bradycardia or dyspnea which provoked a change in the therapeutic regimen. It is known that concomitant use of BB with the substrates of CYP2D6 may lead to clinically-significant drug–drug interactions which may thus potentiate the adverse cardiovascular effects of topically administered timolol [39]. Knowledge of interactions between timolol and other drugs is important, especially as many glaucoma patients are elderly and often use concomitant drugs, which may influence metabolic enzyme activities. 

Our study presents some limitations. Due to the nature of real life study of our research, we have included patients treated with a combination of hypotensive drugs. To be strict, this kind of studies should be performed only with patients in monotherapy to eliminate potential drug–drug interactions between the evaluated drugs and previously administered or concomitant medication. Considering the real needs of this population, pharmacokinetics and pharmacodynamics interactions leading to a personal response could be expected from the different previous treatments or the actual combined treatment. Several publications have analyzed resistance to these types of drugs. Besides, the problem of non-adherence in glaucoma is very important and can interfere with these types of studies [40]. 

Another limitation is related to ethnicity. All patients involved in this study were Caucasians. There is a need to validate this data in patients with different ethnicities and frequency for the evaluated genetic variants to better evaluate the association between the genetic variants and response to treatment. It is of special interest, particularly for those heterozygous carriers, to combine the genotype with the analysis of phenotype in an attempt to better select and adjust the doses in each patient. Further studies should consider drug exposure, the genotype and its association with target concentrations achievement, and clinical outcomes improvement.

Our study has clinical utility as it approaches personalized medicine. Currently, we do not have known pharmacogenetic criteria to establish the best drug to prescribe a therapeutic regimen for a given patient; however, knowing these markers before starting treatment can help to select a drug that achieves greater effectiveness and less toxicity. Moreover, larger glaucoma pharmacogenetic studies on this topic are needed in order to be able to design precise therapeutic regimens for patients and modulate the dose of the prescribed drugs.

## 4. Materials and Methods

### 4.1. Participants and Sample Selection

A prospective, observational, and consecutive clinical case study was conducted among 72 patients affected with glaucoma or OHT who were treated with topical BB and/or PGA. In the studied population, 67 patients were treated in both eyes and 5 patients only in one eye. Consecutive patients were recruited from September 2020 to April 2021 in outpatient clinic rooms of the Institut Clínic de Oftalmologia (ICOF), Hospital Clínic de Barcelona. Genotype analysis was carried out in the Pharmacology and Toxicology Unit of Biochemistry and Molecular Genetics Department of the same hospital. The study was conducted in accordance with the tenets of the Declaration of Helsinki. The protocol of this research was approved by the local ethics committee. Informed consent was obtained from each participant before the study.

### 4.2. Inclusion Criteria

All patients were over 18 years old, diagnosed with glaucoma or OHT, and treated with BB and/or PGA alone or combined, regardless of the duration and dose of the treatment. As well, in some cases, other types of ocular hypotensive treatments were also used in combination with the studied drugs. Patients who were interested in participating and willing to give us a blood sample were included. 

### 4.3. Exclusion Criteria

Patients without hypotensive treatment, patients with history of any kind of intraocular surgery in the last 3 months, patients with intercurrent intraocular acute condition requiring other types of drugs (i.e., steroids, antibiotics, intravitreal therapy), or patients with eyes in phthisis bulbi condition were excluded from the analysis. 

### 4.4. Methodology

Recruited patients underwent complete ophthalmology exploration in order to collect ocular data relative to clinical response to the studied drugs. Blood samples were drawn for analysis by means of real time PCR assays of 5 SNPs, already described by their relationship with the response to treatment. The study employed two SNPs of the *PTGFR* gene, rs3766355 and rs3753380, related to the response to prostaglandin analogues, and three SNPs related to the response to beta-blockers; one with *ADBR2* rs1042714 and two with *CYP2D6* rs16947 and 769,258 were analyzed. 

### 4.5. Analysis of Genotypes

Genomic DNA was extracted from peripheral whole blood of the patients included in the study. For genotyping, an allelic discrimination reaction was performed using specific Taqman Custom plating SNP of Applied biosystems (reference 4462782) with lyophilized probes and primers with the following polymorphisms: *PTGFR* rs3753380 (C___1686003_10), *PTGFR* rs3766355 (C__25807762_10), *ADRB2* rs1042714 (C___2084765_20), *CYP2D6* rs16947 (C__27102425_10), and *CYP2D6* rs35742686 (C__27102444_F0) on a ViiA™ 7 Real-Time PCR System (Applied Biosystems). Procedure: 30 µL of DNA at 4 ng/µL from each sample or control is mixed with 30 µL of TaqPath™ ProAMP™ Master Mix (Applied biosystems with ref. A30865) in an eppendorf tube (a total of 60 µL for each sample or control). Then, 10 µL of each sample or control is dosed to the 5 wells of each SniP of the personalized plate, leaving a final concentration of 2 ng/µL. The plate is covered with adhesive (optical film). A centrifuge pulse is given (8″–10″ at 1800 rpm). Finally, the plate is analyzed in the ViiA 7 equipment. Thermal cycler conditions are summarized in Table 4. Quality analysis: Results were analyzed with Taqman Genotyper software after determination in the Viia7 Real-time system (Thermofisher Scientific).

The data obtained from the ophthalmological medical records and the data from the results of the PCR analysis were collected. We obtained the following data from the medical records: demographic features, baseline IOP (without treatment), treatments used, treated IOP, medication side effects, diagnosis (closed angle glaucoma, open angle glaucoma, secondary glaucoma, or ocular hypertension), visual field mean deviation (MD), previous ocular interventions, and systemic concomitant medication treatments.

As baseline IOP (bIOP), we chose the value of the IOP measured without any treatment or after glaucoma surgery if the patient required additional medication. The type of treatment prescribed was collected as indicated for each patient, and the treated IOP was measured at follow-up visits after a certain period of time with a specific treatment when an inappropriate IOP was observed and a change in medication or surgery was required, or when, on the contrary, the pressure was well controlled with a specific drug.

Treated IOP was recorded not before three months of treatment with the studied drug, as some patients are known to be late responders to the commonly-used anti glaucoma drugs [41]. 

The treatment information was classified into six subgroups depending on the combination of treatments: eyes with betablocker treatment alone, eyes with prostaglandin analogues alone, eyes with betablockers and prostaglandin analogues, eyes with betablocker and other treatment, eyes with prostaglandin analogues and other treatment, and, finally, eyes with betablockers, prostaglandin analogues, and other concomitant treatment.

A lack of response was considered to be when the rate of IOP decrease did not reach the known expected hypotensive levels for each drug, namely 25–35% and 20–25% for PGAs and BBs, respectively [4,8,13]. 

About previous ocular surgery, we collected data of all ophthalmological surgical interventions performed and categorized the patients into three subgroups: one without any kind of eye surgery; a second group with some kind of glaucoma surgery (filtering surgery, minimally invasive surgery, drainage devices, etc.); and a third group with other kind of ophthalmological non-glaucoma surgery (cataract surgery or retinal detachment, among others). In the same way, we collected information about all concomitant systemic medications (antilipidemic, antihypertensive, and antidepressant, among others) and searched for information on their possible interaction with CYP2D6 to classify patients into two subgroups: those who were taking a drug that interacted with CYP2 D6 and those who were not.

In reference to side effects, we searched in the medical records for some annotation about adverse drug effects, and recorded the symptoms; bradycardia, dyspnea, dizziness, or allergy. We divided the patients into two subgroups according to the presence or not of those side effects.

### 4.6. Data Analysis

Data were analyzed using SPSS IBM statistics version 27 and 28. For demographic data such as sex, age, and eye laterality and for clinical data such as glaucoma diagnosis, surgery, ocular treatment, adverse drug effect, or systemic treatment, frequency tables were made to study the sample data, obtaining the mean, median, and standard deviation of these. Also, for frequency of the SNP variants (heterozygous, homozygous, and wildtype), bIOP, tIOP, and visual field MD, we obtained frequency tables.

We assessed the quantitative data—baseline IOP, treated IOP, visual field MD—and we compared our sample with a theoretical normality curve through the Kolmogorov–Smirnov test with the Lilliefors and Shapiro–Wilks correction. Our data were significantly different to the theoretical normality and, for this reason, we analyzed them using the Kruskal–Wallis test.

Statistical significance was considered when the *p* value was less than 0.05.

We included both eyes of each patient in the study when available, although separately, as glaucoma is an asymmetrical illness, and topical hypotensive treatments are often different between both eyes.

Finally, we performed two additional analyses. The first one was intended to verify if age was a determining factor for drug response, for which patients were divided into two subgroups according to age (less than 60 or more than 60 years old). The second one considered two different subgroups of patients, whether they were being treated with prostanoids or with prostaglandin analogues, to verify if the difference in structure could explain a different clinical response between both types of drugs. 

## 5. Conclusions

Glaucoma is a potentially blinding disease. In some cases, the rate of progression of the disease is particularly fast, making it mandatory to establish an early effective therapeutic regimen. Further, especially when young patients are affected, the disease presents with particular severity and empiric treatments can sometimes result in unexpected visual loss. 

This is the first study to our knowledge that looks at the pharmacogenetic background of a glaucoma or OHT patient’s population, their clinical characteristics, and the response to treatment with prostaglandin analogues and beta-blockers from both the pharmacodynamic and pharmacokinetic aspects. In our study, a statistically significant concordance was observed between the mutated alleles in the SNPs rs3766355 *PTFGR* gene and higher treated IOP, and also between mutated alleles in the SNPs rs3766355 and rs3753380 of the *PTFGR* gene, and worse MD in visual fields. These preliminary results suggest the potential of pharmacogenetic-based treatment to improve the clinical outcomes of patients with glaucoma.

## Figures and Tables

**Table 1 ijms-24-02093-t001:** Demographic and clinical features.

Features	Frequency	Percentage
**Sex**		
Female	40	55.5
Male	32	44.5
**Eye laterality**		
Right	68	48.9
Left	71	51.1
**Eye Diagnosis**		
Primary Open Angle Glaucoma	92	66.2
Angle Closure Glaucoma	13	9.4
Ocular Hypertension	18	12.9
Secondary Glaucoma	16	11.5
**Eye Surgery Records**		
No surgery	63	45.3
Glaucoma surgery	57	41
Other surgery	19	13.7
**Hypotensive Treatment per Eye**		
Beta-blockers	9	6.5
Prostaglandin analogues	29	20.9
Beta-blockers + Prostaglandin analogues	24	17.3
Beta-blockers + other	18	12.9
Prostaglandin analogues + other	2	1.4
Beta-blockers + Prostaglandin analogues + other	57	41
**Ocular medication Side Effect**		
No	66	91.7
Yes	6	8.3
**Systemic Treatment**		
Metabolized by CYP2D6	17	23.6
Not metabolized by CYP2D6	55	76.4

**Table 2 ijms-24-02093-t002:** IOP and Visual field data.

	Age	MD Visual Field (dB)	Baseline IOP mmHg	Treated IOP mmHg	Rate of IOP Variation
**Mean**	63.99	−7.59	26.55	21.01	−16.62
**Median**	65	−3.58	24	20	−14.28
**Deviation**	11.99	8.63	9.198	5.543	25.94

MD mean defect, IOP intraocular pressure; dB decibels.

**Table 3 ijms-24-02093-t003:** IOP and visual field parameters according to the genetic profile.

	Baseline IOP ± SD	Kruskal-Wallis Test (*p*)	Treated IOP ± SD	Kruskal-Wallis Test (*p*)	MD Visual Field (dB) ± SD	Kruskal-Wallis Test (*p*)
***PTGFR* rs3766355**						
WT	23 ± 3.625	*p* = 0.498	16 ± 2.16	*p* = 0.031	−2.07 ± 2.20883	*p* = 0.009
HT	26.58 ± 8.465		21.70 ± 3.486		−3.87 ± 4.0248	
HM	26.97 ± 8.445		20.98 ± 6.125		−9.37 ± 9.51008	
***PTGFR* rs3753380**						
WT	26.67 ± 6.799	*p* = 0.232	21.38 ± 4.744	*p* = 0.119	−6.10 ± 8.67477	*p* = 0.017
HT	26.88 ± 9.271		20.98 ± 6.341		−9.02 ± 8.63445	
HM	22.20 ± 10.640		17.50 ± 3.082		−9.51 ± 7.44773	
***ADBR2* rs1042714**						
WT	25.26 ± 8.128	*p* = 0.200	20.58 ± 4.515	*p* = 0.780	−9.48 ± 10.70494	*p* = 0.733
HT	28.22 ± 8.115		21.44 ± 20.5		−7.39 ± 7.93112	
HM	26.59 ± 9.449		20.53 ± 4.69		−6.22 ± 7.13030	
***CYP2D6* rs16947**						
WT	28.44 ± 10.374	*p* = 0.109	21.08 ± 6.253	*p* = 0.143	−6.79 ± 8.48020	*p* = 0.311
HT	24.38 ± 5.034		21.36 ± 5.063		−8.58 ± 8.82571	
HM	29.77 ± 9.506		18.88 ± 4.884		−7.16 ± 8.87592	
***CYP2D6* rs769258**						
WT	26.79 ± 8.631	*p* = 0.753	20.62 ± 5.465	*p* = 0.181	−7.86 ± 8.76595	*p* = 0.311
HT	22.37 ± 5.354		22.74 ± 5.886		−6.37 ± 8.17390	

WT: wildtype; HT heterozygous; HM homozygous; MD mean defect, IOP; intraocular pressure; dB decibels; SD standard deviation.

**Table 4 ijms-24-02093-t004:** PCR Thermal cycler conditions.

Step	Stage	Time	Temperature
Inicial steps	Hold	10 min	95 °C
Denature	40 cycles	15 s	95 °C
Anneal/Extend	40 cycles	60 s	60 °C

## Data Availability

Our data are unavailable due to privacy and ethical restrictions.

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
