# Peer review of "Genotype and Phenotype Influence the Personal Response to Prostaglandin Analogues and Beta-Blockers in Spanish Glaucoma and Ocular Hypertension Patients"

_ijms, 2023, doi:10.3390/ijms24032093_

Round 1

Author Response

Thank you very much for your useful and constructive suggestions.  Please see the attachment

Reviewer 2 Report

The manuscript entitled “Genotype and Phenotype influence the personal response to prostaglandin analogues and beta-blockers in Spanish glaucoma and ocular hypertension patients” is a study about the response of different drugs in patients with glaucoma and ocular hypertension patients.

The following points are addressed:

-   The title has grammatical errors, is “analogues” not “analogs”.

-       If acronyms appear for the first time in the abstract, identify their meaning. Line 50, bIOP and tIOP.

-       As geneticists, you should know that genes should be written in italics so you can distinguish them from protein action.

-       Line 48, line 49, line 209, line 254 writing error. Don't start a sentence with a number, start with a letter.

-       The introduction should be generic showing the context from which the research arises, it is not correct to say, "we use different groups of drugs to lower IOP" Line 87.

-       Line 96 little justified, add more references.

-       The introduction is not ordered, you should organize it, talking first about the treatment for this disease, then about its effect, response and finally introducing genetics as a tool to improve the response.

-       Line 106 PTGFR: which means, explain the acronym of this protein.

-       Line 107 “Various single nucleotide polymorphisms (SNPs) have been reported to be associated with better or worse response to treatment. [7,8,9,10]”. Which? If the result is controversial, why analyse it?

-       Line 136-139, no references.

-       Line 140, what is OHT?

-       Line 145: Using the term "eye" may be inappropriate, better to specify the eyes of patients. Indicate that in some patients both eyes are treated and in others only 1.

-       Line 154: Lack of information: duration of treatment, at what dose...

-       Line 166: Why are only two CYP2D6 polymorphisms analysed? With this information it is not possible to know if it is a poor or fast metabolizer. This gene is highly polymorphic, and to obtain the phenotype, its complete genotype must be studied.

-       Line 183: It is not stated how adverse effects are measured.

-       Line 188: why it is used two versions of SPSS?

-       Line 202: Justify why the age of 60 years is chosen as the reference value to make two subgroups.

-       The relationship between the genotype and the eyes does not make sense. Regardless of the treated eye, the patient will have the same genotype. The association must be made with the patient, not with the eye.

-       Figure 1 is unnecessary; it does not contribute anything if the same data is indicated in the text.

-       Line 227, it is not explained the most important results. This is the most relevant part of the study and nothing about it is explained in the results part.

-       Results: Nothing is disclosed about the association of genes with clinical values. The adverse effects are not mentioned either, why were they measured if later no information is provided in this regard?

-       Discussion: In this paper, the discussion is lacking, it can be explained more fully and make a reasonable explanation. Line 252-255 Nothing is said, not does it contribute anything in relation to what was obtained with the study or in previous studies.

There are almost no references, and all those indicated in the introduction are indicated in the discussion. It is poor in terms of scientific relevance. The analysis of the results is defective, and the discussion is very poor.

Author Response

Thank you for your useful suggestions. Please see the attachment

Round 2

Reviewer 2 Report

Unfortunately, despite correcting certain comments, not all of them have been corrected properly.

New parts of text without references have been added: line 90-91, 92-93, 102, 116-117, 138-140. The review of the literature is somewhat selective also, as it only emphasizes the positive studies on the relationship between specific genotypes and response. The discussion is still poor in terms of scientific relevance.

Regarding CYP2D6, its analysis is not well justified. The authors indicate in the “Response to Reviewer 2 Comments” that: “two polymorphisms were selected according to articles published in the world literature” but these references do not appear in the manuscript. The references do not focus on treatment with glaucoma and beta blockers.

Again the analysis of the results of lines 323 to 331 is not adequate. The genotypic frequency cannot be per eye, it must be per patient, although later the association between the pharmacological response obtained in each eye with the polymorphisms is to be analyzed.

The measurement of adverse effects is not sufficiently robust and reliable. Specific evaluated evidence regarding the adverse effects of pharmacological treatment that is validated must be used. With the data provided, selected signs and symptoms may or may not be associated with that drug; and it is not reported whether the adverse effects persist or have disappeared.

The scientific value of this manuscript is low and clinical relevance is not given.

Author Response

Thank you very much for your useful suggestions. 

Round 3

Reviewer 2 Report

Despite the changes provided, the quality of the article has not been improved. A lack of scientific writing has been observed. Readers could not find the key messages from the article. Authors’ modifications are imprecise and poorly elaborated. There is a great deal of typographic and redaction mistakes. In addition, proposed changes do not correspond to quality standards of the journal.

I therefore don’t recommend publication.